# The Impact of COVID-19 on Access to Resources among Individuals Experiencing Homelessness and Traumatic Brain Injury

Stephanie Chassman [1,*], Blair Bacon [2], Sara Chaparro Rucobo [3], Grace Sasser [1], Katie Calhoun [1], Emily Goodwin [4], Kim Gorgens [3] and Daniel Brisson [1]

1    Graduate School of Social Work, University of Denver, 2148 S. High St., Denver, CO 80208, USA
2    Department of Psychiatry, University of Colorado School of Medicine, Aurora, CO 80045, USA
3    Graduate School of Professional Psychology, University of Denver, Denver, CO 80210, USA
4    Rosemead School of Psychology, Biola University, La Mirada, CA 90638, USA
*    Correspondence: stephanie.chassman@du.edu

**Abstract:** The rates of traumatic brain injury (TBI) are higher among individuals experiencing homelessness compared with the general population. Individuals experiencing homelessness and a TBI may experience barriers to care. COVID-19 may have further impacted access to basic resources, such as food, shelter, and transportation for individuals experiencing homelessness. This study aimed to answer the following research question: What is the impact of COVID-19 on access to resources among individuals experiencing homelessness and TBI? A cross-sectional study design and purposive sampling were utilized to interview 38 English-speaking adults experiencing homelessness and who had sustained a TBI (ages 21–73) in one Colorado city. Qualitative questions related to the impact of COVID-19 were asked and qualitative analysis was used to analyze the responses. Three primary themes emerged regarding the types of resources that were restricted by COVID-19: basic/biological needs, financial needs, and a lack of connection. COVID-19 has shown the social work field the need for continued innovation and better practice standards for individuals who are not housed. For those living with a reported TBI history and experiencing homelessness, COVID-19 made it difficult to access basic services for survival.

**Keywords:** homelessness; traumatic brain injury; COVID-19





## 1. Introduction

### 1.1. Overview of Homelessness

In 2020, more than 580,000 people experienced homelessness in the United States (U.S.) [1]. The National Alliance to End Homelessness (2021) explains that experiencing homelessness is accompanied by risk factors for poor health, such as uninhabitable living conditions, violence, substance use, decreased access to health care, and health issues in general, including, but not limited to, traumatic brain injury (TBI).

### 1.2. Overview of TBI

A TBI is a head injury that disrupts brain functioning, often caused by an external force [2,3]. There are various potential causes of TBI, including exposure to violence (e.g., assault), falls, motor vehicle accidents, substance-use related accidents, and sports [4]. Several studies suggest that if one head injury has occurred, the likelihood of subsequent head injuries increases, with repeat head injury found in 45–60% of cases [5–7].

TBI is associated with many negative outcomes, including lifelong cognitive impairments that harm an individual's memory processing, attention, communication, and executive functioning [8]. A TBI can meaningfully impact a person's capability to self-regulate, plan, and organize. A TBI affects a person's judgment, reasoning, and problem-solving

ability; changes in one's emotions and behaviors, as well as one's ability to regulate themselves, are also common [9]. A recent systematic review and meta-analysis found that a history of TBI was associated with poor physical and mental health, suicidality, memory concerns, higher health service use and criminal justice system involvement [10]. The risk factors associated with TBI may also lead someone to experience housing instability.

### 1.3. Intersection of TBI and Homelessness

The rates of TBI are significantly higher among individuals experiencing homelessness than the general population. Research has shown that more than half of individuals experiencing homelessness have sustained a TBI at some point in their lifetime compared to 2–8.5% of housed individuals [3,5,7,10,11]. Individuals experiencing homelessness are at a disproportionately high risk of sustaining a TBI. This is in part because individuals experiencing homelessness are more likely to be the victims of assaults, have a higher propensity for risk-taking, and have higher rates of substance use [12,13]. The relationship between TBI and homelessness is not well understood. In order to better understand the intersection of TBI and homelessness, this study will utilize Maslow's hierarchy of needs as a theoretical framework.

### 1.4. Impact of COVID-19

Individuals experiencing homelessness may be at higher risk of contracting COVID-19 and experience more difficulty recovering from the virus, as they often have higher rates of physical health challenges compared to their housed counterparts [8,9]. In addition to potentially being at high risk of contracting COVID-19, access to care for people experiencing homelessness was further complicated in the pandemic as clinics deferred in-person care to telehealth visits [14]. While using telehealth has expanded healthcare access for some, it disadvantages people experiencing homelessness as it requires a reliable internet connection and particular technology, which people experiencing homelessness may not be able to access. Many businesses and libraries were closed to the public during the pandemic, further limiting access to reliable internet connection. Additionally, people experiencing homelessness may be less likely to have a smartphone due to the costs. A study by Raven et al. (2018) found that among 350 adults experiencing homelessness in Oakland, CA, more than 72% of the participants had access to a mobile phone, most of which were not smart phones. Just over half (55%) reported using the internet, whether on a phone, in a library, day shelter, friend or family's home, a coffee shop or restaurant, social service agency, motel/hotel, or church (Raven et al., 2018). Participants with executive function impairment were even less likely to have access to a mobile phone [15].

COVID-19 also impacted access to basic resources, such as food, shelter, and transportation. Feeding America (2021) estimated that between 10.5 and 17 million more people in the U.S. will experience food insecurity during and after the pandemic, and reported that food banks are serving 55% more people than before the pandemic. Particularly in the beginning months of the pandemic, hundreds of pantries closed after older volunteers stayed home and staff called in sick [16]. An increased demand for food, as well as an initial decline in food bank operating capacity, likely further impaired food access for people experiencing homelessness and living with a TBI.

Access to homeless shelters was also affected by the COVID-19 pandemic. A Department of Housing and Urban Development (HUD; 2021) report explains that for the first time since the Point in Time data collection began, more individuals experiencing homelessness were unsheltered than sheltered. The number of unsheltered individuals increased by 7% in 2020 [17]. Two potential reasons for the decline in shelter stays are decreased shelter capacity and the risk of COVID-19 transmission. To decrease the risk of transmission, shelter operators were tasked with implementing social distancing measures, thus reducing their capacity [18]. Additionally, Perri and colleagues (2020) explain that shelters may be a dangerous place regarding COVID-19 transmission because of the

shared spaces, amount of people, difficulty implementing physical distancing measures, and clientele turnover [19].

Transportation access for people experiencing homelessness during the pandemic has been impacted by a decrease in public transit service and diminished ability to receive rides from others due to the risk of COVID-19 transmission [14]. Transportation is essential for accessing basic needs, such as food, shelter, and healthcare. Individuals with disabilities, such as a TBI, generally need access to healthcare services more often and are more likely to have delayed access to the necessary care or to have gone without it compared to those without disabilities [20]. Individuals with disabilities are also more likely to utilize transportation, such as public transportation, that exposes them to others [14].

*1.5. Literature Review*

1.5.1. Theoretical Framework

Maslow's hierarchy of needs (1943) is a theory that incorporates a five-tier model of human needs. These needs are organized as hierarchical levels, where the baseline needs to be satisfied before needs higher up can be attained. The five tiers are as follows [21]: beginning with base needs, moving to psychological needs, and leading to self-actualization. Physiological needs are biological requirements for human survival, which include food, water, warmth, and rest. Safety needs include security and safety. Physiological and safety needs are considered basic needs. Belongingness and love needs refer to intimate relationships and friends. Esteem needs include prestige and feelings of accomplishment. These two are considered psychological needs. Lastly, the self-actualization tier includes achieving one's full potential and creative activities. This tier is considered to be self-fulfillment needs.

When utilizing Maslow's theory, one needs to examine homelessness as a social problem that affects various dimensions of human needs. Limited access to services, in general, has the potential to exacerbate homelessness. Additionally, limited access to social and medical services may complicate treatment for a TBI. Homeless service organizations have warned that COVID-19 could cause catastrophic harm to homeless communities due to the absence of a coordinated plan for these often-overlooked individuals [22]. Individuals experiencing homelessness often have more unmet needs compared with the general population [23], potentially making it more difficult to progress up the hierarchy of needs. Considering that individuals experiencing homelessness and a TBI are more likely to have a greater extent of unmet needs, there is a clear need for programs that are equipped to respond in times of crisis. Utilizing Maslow's hierarchy of needs to better examine the relationship between TBI and homelessness, this study will review the existing barriers to care in terms of access for people experiencing homelessness and TBI.

1.5.2. Barriers to Care

Health Care

Individuals experiencing homelessness and living with a TBI may experience additional barriers to care due to the consequences of a TBI. Memory loss, a common consequence of a TBI [8], may affect one's ability to remember appointments. People experiencing homelessness who do not have a mailing address may have a difficulties receiving important documents related to their healthcare and health insurance. Research has shown health insurance to be a key factor in obtaining healthcare [23]. Individuals experiencing homelessness may lack access to health insurance, which further impacts their ability to receive vitally important medical care for diagnosing and treating a TBI, such as neuroimaging and clinical documentation. One study found that among 134 people experiencing homelessness, 70% were unaware of their potential Medicaid eligibility [24]. The researchers also found that people experiencing homelessness were less likely to have knowledge of the Affordable Care Act than housed participants [24].

Financial Barriers

In addition to the financial hardship that may have led to homelessness, individuals experiencing homelessness and living with a TBI may face additional financial barriers. TBIs can affect one's cognitive functioning, causing confusion, difficulty with memory and making decisions, planning, and organization [25], which may make it difficult to obtain and maintain employment. Research suggests that moderate to severe TBIs can also affect financial decision-making skills, such as managing a checking account, paying bills, and other more complex tasks, particularly within the first six months of sustaining a TBI [26]. These difficulties can exacerbate existing financial hardship for people experiencing homelessness and living with a TBI.

Identification

Another barrier to care that people experiencing homelessness often encounter is obtaining and accessing services that require personal identification [27]. Without personal identification, it may be difficult to access resources, such as health care, housing, banking services, employment, and public benefits. It is also often challenging to access emergency food services, such as food banks, without personal identification and proof of residence [27]. This issue is particularly pronounced for people experiencing homelessness, who may be more susceptible to having their belongings stolen or lost. Sanders and colleagues (2020) explain that people experiencing homelessness are less likely to have the resources to obtain new personal identification, such as funds for fees and an understanding of the process and forms. A TBI may further impact the barriers to health insurance, personal identification, and managing finances due to the negative consequences of a TBI. Upon examining the barriers to care for people experiencing homelessness and living with a TBI, it becomes clear that the intersection of TBI and homelessness is multifaceted. The above-mentioned barriers to care were likely exacerbated during the COVID-19 pandemic.

*1.6. The Current Study*

Based on a universal lack of access to resources during COVID-19 and the unique vulnerabilities of persons with a history of brain injury who are homeless, this study aims to answer the following research question: What is the impact of COVID-19 on access to resources among individuals experiencing homelessness and a TBI?

## 2. Materials and Methods

The qualitative components of the data set examined the impact that COVID-19 had on access to resources among individuals experiencing homelessness and a TBI among 38 participants from one study site (Colorado Springs).

A cross-sectional study design and purposive sample were utilized to interview a total of 56 English-speaking adults (ages 21–73). The University of Denver (Center for Housing and Homelessness Research and the Graduate School of Professional Psychology) worked with community partners to recruit potential participants.

Data collectors utilized an eligibility screener to assess whether the participants were over 18 years old and experiencing homelessness or in unstable housing; if participants were eligible, they were asked to give written consent. The Institutional Review Board (IRB) at the University of Denver approved all of the study procedures prior to the data collection.

During data collection, the participants had the option to skip any questions and seek support from trained staff if needed. The survey took approximately 25 min to complete. Participants were given a $15 gift card as compensation.

*2.1. Measures*

First, the participants' date of birth, age, and military service (yes, active; yes, veteran; no) information was collected. Then, additional sociodemographic information was collected, including: gender identity [male, female, other (specify)], sexual orientation (heterosexual or straight, gay, lesbian, bisexual, not listed above), race (American

Indian/Alaska Native, Asian, Native Hawaiian or Other Pacific Islander, Black or African American, White, Hispanic, more than one race, unknown/not reported), and level of education: less than a high school diploma; high school degree or equivalent; Associate's degree; Bachelor's degree; Master's degree; Doctorate; other).

Two standardized measures were used to assess the participants' homelessness status and TBI history. The Vulnerability Index—Service Prioritization Decision Assistance Tool (VI-SPDAT) was used to assess homelessness and the Ohio State University TBI Identification Method (OSU TBI-ID) was used to assess TBI history.

### 2.2. VI-SPDAT

The VI-SPDAT [28] was used to assess the history of housing and homelessness, risk behavior, socialization, and daily functioning and wellness.

### 2.3. OSU TBI-ID

The OSU TBI-ID [29] was used to collect information on TBI. This self-report screener assesses whether participants have a significant history of TBI, if they reported a "first" TBI with loss of consciousness (LOC) before age 15, a "worst" TBI with LOC longer than 30 min, or a "multiple" TBI "a period where three or more blows to the head caused altered consciousness OR two or more TBIs with LOC within a 3-month period" [30]. Scores of first, worst, or multiple were utilized and if a participant scored any of the criteria, they were scored as having a TBI (1 = yes, 0 = no).

### 2.4. COVID-19

COVID-19 related questions were included in Colorado Springs among 56 total participants. COVID-19 related questions were not included at the first data collection site (Fort Collins) because the data collection took place before COVID-19.

The questions related to the impact of COVID-19 included quantitative questions, where participants answered yes or no, including: Have you tested positive for COVID-19? Has having COVID-19 resulted in other medical conditions to worsen? Did you experience homelessness for the first time due to COVID-19? Are you currently experiencing homelessness due to COVID-19? In addition, a qualitative component where participants were asked: Has COVID-19 restricted your access to resources (e.g., medical, food pantries, shelters, housing, etc.)? The participants were asked to specify and expand upon which specific resources were restricted during COVID-19.

### 2.5. Analytic Approach

The study data were collected using the REDCap (Research Electronic Data Capture; [31]) electronic data capture tools hosted at The University of Denver. Listwise deletion was utilized for missing data because less than 10% of the data were missing [32]. Quantitative analysis was used for analyzing the sociodemographic characteristics, VI-SPDAT, the OSU TBI-ID, and the quantitative portion of the COVID-19 questions.

Qualitative analysis was used to analyze the open-ended COVID-19 related questions: Has COVID restricted your access to resources (e.g., medical, food pantries, shelters, housing, etc.)? Specify which resources were restricted. The COVID-19-related responses were analyzed using qualitative methods, including open, first, and second cycle coding [33]. The researchers first used open coding to document the initial reactions to the data. Once the researchers completed the open coding independently, we came together to discuss and reach an agreement on the codebook. We then used code mapping to categorize the codes, followed by axial coding to reduce redundancy. This allowed the researchers to see emerging themes.

## 3. Results

A total of 56 individuals experiencing homelessness completed the survey on October 2nd, 2020. Among the 56 participants, 38 had a TBI. The quantitative descriptive charac-

teristics, homelessness-related variables, and TBI-related variables for the 38 participants experiencing homelessness with a TBI are reported in Table 1.

**Table 1.** Descriptive Characteristics of Participants.

| Descriptive Characteristics of Participants (N = 38) | |
| --- | --- |
| | **n (%) or M (SD)** |
| Age (years) | 48.7 (11.6) |
| Gender | |
| Male | 29 (76.3) |
| Female | 8 (21.1) |
| Gender minority | 1 (2.6) |
| Sexual Orientation | |
| Heterosexual | 35 (92.1) |
| Bisexual | 2 (5.3) |
| Gay | 1 (2.6) |
| Race and Ethnicity | |
| White | 22 (57.9) |
| Latinx | 3 (7.9) |
| Multiracial | 3 (7.9) |
| Black | 4 (10.5) |
| American Indian/Alaska Native | 5 (13.2) |
| Unknown/Not Reported | 1 (2.6) |
| Education | |
| Less than High school diploma | 3 (7.9) |
| High school education or equivalent | 19 (50) |
| Associate degree | 4 (10.5) |
| Bachelor's Degree | 5 (13.2) |
| Master's Degree | 3 (7.9) |
| Other | 4 (10.5) |
| Veteran (1 = yes) | 9 (23.7) |
| Homelessness Variables | |
| Sleep in shelters | 14 (36.8) |
| Sleep outside | 16 (42.1) |
| Other | 5 (13.2) |
| Transitional housing | 1 (2.6) |
| Safe haven | 2 (5.3) |
| Time without stable housing | 4.3 yrs (9.2) |
| No. episodes of homelessness | 3.6 (6) |
| Age of first experience homelessness | 32.4 (14.8) |
| TBI Variables | |
| Head injury as a barrier to housing | 11 (28.9) |
| "Worst" injury | 24 (63.2) |
| "First" injury | 11 (28.9) |
| "Multiple" injury | 22 (57.9) |

### 3.1. Descriptive Characteristics

Out of the 38 participants from Colorado Springs who had a TBI, 76% identified as male, followed by 21% who identified as female, and 2% who identified as neither. Regarding sexual orientation, most of the participants identified as heterosexual (92%), while 5% identified as bisexual, and 2% as gay. Most of the participants identified as White (58%), followed by American Indian/Alaska Native (13%). The mean age was 48.7 (SD = 11.6), most of the participants did not identify as a Veteran (76%). Half of the participants had earned a high school degree or equivalent (50%), followed by a bachelor's degree (13%), and an associate's degree (10%).

### 3.2. Homelessness Variables

Regarding the homelessness characteristics, 42% of the participants claimed they slept outdoors most frequently, followed by 37% who slept in shelters, 13% slept in other

locations not listed, 5% slept in safe havens, and 2% slept in transitional housing most frequently. On average, the participants had not lived in permanent stable housing in 4.3 years (SD 9.2) and had experienced 3.6 different episodes of homelessness (SD = 6). The mean age at which the participants first experienced homelessness was 32.4 years old (SD =14.8).

### 3.3. Brain Injury

When examining the brain injury-related variables, the participants reported that living with a brain injury was a barrier to housing stability. Specifically, 29% of participants claimed that a past brain injury had served as a barrier to maintain housing or was the reason a participant was evicted from their apartment or shelter program. The OSU TBI-ID screening revealed that 63% of the participants reported at least one head injury with a LOC for more than 30 min (worst). In addition, 28% of the participants reported experiencing a TBI with LOC before the age of 15 (first). Additionally, 58% of the participants reported experiencing either three or more head injuries resulting in an altered state, meaning a LOC or being dazed or two or more TBIs with LOC within a three-month period (multiple).

### 3.4. COVID-19 and Homelessness

Several participants reported that COVID-19 had impacted their homelessness status, as well as access to resources. While no participants claimed they had tested positive for COVID-19, 71% of persons with a reported TBI history said that COVID-19 had restricted their access to resources (e.g., medical, food pantries, shelters, housing, etc.) compared to 50% of participants with no reported history of TBI. Additionally, 13% of persons with a reported TBI history said they experienced homelessness for the first time due to COVID-19, compared to 11% of participants with no reported history of TBI. Moreover, 36% of persons with a reported TBI history said they were experiencing homelessness currently due to COVID-19 compared to 22% of participants with no reported history of TBI.

### 3.5. Emergent Themes

While four participants reported that "everything is closed," three primary themes emerged regarding the types of resources that were restricted by COVID-19: basic/biological needs, financial needs, and lack of connection. Subthemes within these larger themes are described below.

#### 3.5.1. Basic, Biological Needs: Food, Shelter and Hygiene

Food, shelter, and hygiene resources were all categorized as basic needs. A lack of access to food resources, including food pantries and organizations that offer meals, were common responses among the participants. Multiple participants reported that "food pantries" were closed. Additionally, participants reported that "shelters" were closed, and they lacked access to "housing" resources. Another basic need that was impacted was hygiene resources. Public places, such as churches, where participants could take a shower were shut down. One participant claimed, "It is harder to find a place to take a shower." Other basic needs that were restricted include access to health care, medical resources, and clothing, as one participant reported, "I can't access a psychiatrist due to telehealth and not having a computer."

#### 3.5.2. Financial Needs

A lack of access to financial needs came up as a central theme. The participants reported lacking access to income and money services, the bank, mail, employment, government services including the DMV to obtain identification, and public transportation. As one participant reported, "The mail is backed up and slow to receive Disability" and another, "I can't get through to Social Security or go to the office." Another resource that was restricted was access to the stimulus checks sent out for COVID-19 relief. These resources can also be considered basic needs in that, before an individual can consider addressing

basic needs such as food insecurity, they need money to purchase food and a way to travel to the food bank.

### 3.5.3. Lack of Connection

Having a sense of connection is important among individuals experiencing homelessness as a means of exiting homelessness and improving non-housing outcomes [34]. The participants identified sources of social support that were restricted, including "internet access" at libraries as a means to communicate with friends and family, "social gatherings" at food courts and public spaces, church services, and being able to see friends and family. One participant reported that it was "hard to find places to hang out" as many public spaces were closed and unsafe to go to.

## 4. Discussion

The present study examined the impact that COVID-19 had on access to resources among individuals experiencing homelessness and TBI. The results suggest that the overall access to resources was impacted by COVID-19 more so for participants with a reported TBI history (71%) compared to participants with no reported history of TBI (50%), more participants (13%) with a reported TBI history experienced homelessness for the first time compared to participants with no reported history of TBI (11%), and participants with a reported TBI (36%) reported experiencing homelessness currently due to COVID-19 compared to 22% of participants with no reported history of TBI. This further shows the unique vulnerabilities faced by individuals experiencing homelessness with a reported history of TBI. These findings and their implications are discussed next.

### 4.1. TBI

As expected, our results align with previous research that found more than half of individuals experiencing homelessness had a history of TBI at some point in their lifetime [10]. TBI can bring many emotional and social consequences that can add challenges to maintaining housing; additionally, a lack of awareness about TBI may be one of the largest barriers to accessing care among individuals experiencing homelessness [35]. Organizations serving homeless populations may consider screening for a history of TBI as a first step toward achieving client appropriate care.

### 4.2. Homelessness and COVID-19

The results showed that 36% of the participants with a reported TBI history were currently experiencing homelessness due to COVID-19 compared to 22% of participants with no reported history of TBI. Additionally, 13% of participants with a reported TBI history experienced homelessness for the first time due to COVID-19 compared to 11% of participants with no reported history of TBI. In addition, 71% of individuals with a reported TBI history and experiencing homelessness reported restricted access to resources, compared to 50% of participants experiencing homelessness with no reported history of TBI. The themes that emerged from interviews with participants with a reported history of significant brain injury were that the participants experienced a lack of basic needs (food, shelter, and hygiene) and a lack of access to financial needs. Connection was also negatively impacted by COVID-19. Across the country, homeless service providers struggled to respond to the COVID-19 pandemic. Some shelters were forced to reduce their services, restrict admittance, or close entirely in order to follow the public health guidelines and help ensure people's safety [36]. The results of this study highlight the need to open a dialogue to address the unique needs and vulnerabilities of individuals experiencing homelessness and living with a TBI and address planning, responding, and recovery efforts during times of crisis.

### 4.3. Results Examined through Maslow's Hierarchy of Needs

We found that access to resources that help meet one's safety needs, including financial assistance—as described by Maslow (1943), security, employment, resources, health, and property—were also impaired during the COVID-19 pandemic. The participants experienced difficulty in accessing transportation and financial services, which may aid in meeting one's safety needs. A decrease in public transit service and carpooling during the pandemic likely made accessing transportation to essential services, such as food, shelter, employment, and healthcare, even more difficult than before the pandemic. Participants also reported a lack of access to income, employment, and banking services during the pandemic. A lack of access to banking and social security services and slower mail times also impeded the participants' access to income. Without income and/or employment, it is difficult to meet one's biological and safety needs.

The participants also reported a lack of access to connection and social interaction during the pandemic, which connects to Maslow's third tier of love and belonging [21]. The participants referenced a lack of access to places for social infrastructure, such as the library, food courts, restaurant lobbies, and "places to hang out." These locations are typically used for gatherings and/or staying connected to friends and family via the internet and email. The participants also cited a lack of social gatherings and connection, whether through not being able to go to church or to see family and friends. While these needs do not necessarily fall into the categories of biology or safety, connection and social interaction are important for people in general. They are particularly important for people experiencing homelessness and living with a TBI because social support can help connect individuals to resources [37]. Lam and Rosenheck (1999) found that social support was correlated with better health and greater usage of social services among people experiencing homelessness. Additionally, homelessness can be isolating, making a sense of connection even more important for this vulnerable population.

### 4.4. Implications

COVID-19 has illuminated the need for updated policies and better practice standards for individuals who are not housed. For those with a reported history of TBI and experiencing homelessness, COVID-19 made it more difficult to access basic services for survival, as highlighted above. It is evident that additional policies are needed to support people experiencing homelessness and TBIs.

The popular policy, Lifeline, coined "Obama phones," is an example of a policy filling a noticeable need in services, supporting millions of Americans in obtaining phone and internet service [38]. For people experiencing homelessness and a TBI, a heavily subsidized phone plan may help them stay in contact with their social networks, as well as manage various responsibilities. The Federal Communications Commission (FCC) under the Trump administration decreased the budget and capacity of the Lifeline program, impacting the number of people that can access this government program [38]. COVID-19 has demonstrated how critical it is for people to have the ability to maintain contact with their social networks and minimize the effects of quarantine and isolation. Policy recommendations include continued subsidized phone plans and access to reliable internet access in public spaces.

Throughout the pandemic, it was apparent to the local government in Denver, CO, that the current number of congregate shelters in Denver and the surrounding counties were not enough to shelter and quarantine all of the unhoused population [39]. In an effort to provide basic shelter to those in need, local governments utilized existing stadiums and public spaces to transform them into temporary congregate shelters [39]. However, there was still a need for additional sheltering options as the temporary shelters only served a small percentage of the population [39]. Food, another basic need, was more difficult to access due to COVID-19. Low income families were particularly impacted by food insecurity due to lost paychecks or social distancing measures [40]. Medical services transitioned from in-person services to telehealth, requiring consistent access to the internet, computers or

a phone [14]. For people experiencing homelessness, access to the equipment necessary for medical services was limited as day shelters were unable to fully open, libraries were closed, and public access to the internet was restricted. The closing of various community organizations, coupled with many medical services transitioning to telehealth, may have negatively impacted how accessible these necessary provisions were. COVID-19 severely impacted how people experiencing homelessness and a TBI were able to acquire the most basic physiological needs for survival, including shelter, food, and medical care.

Social work practices continue to evolve to match the level of need created by society's circumstances. Community organizations adjusted how they provide services to follow the CDC guidelines and public health protocols [41]. For example, organizations shifted to drive-through services to maintain proper social distancing protocols [42]. It is evident that additional social practice and policy innovation is necessary to better support those living with a TBI and experiencing homelessness. COVID-19 has provided a unique policy window to implement additional social welfare benefits and change the mechanism in which society appropriates funds toward shelters and alternative options. All levels of government administration have taken advantage of this policy opportunity. An example of this is the CARES Act (2020) and the various levels of social welfare programs incorporated into the larger policy. Built into the CARES Act (2020) are additional funds for housing vouchers allocated to city and county governments, funding for 501c(3) organizations, and stimulus checks. It is critical to consider the accessibility of various acts to those who are experiencing homelessness and brain injuries. For instance, three stimulus checks were passed by Congress throughout 2020 and 2021 to provide additional support during the pandemic. However, many individuals that were experiencing homelessness and brain injuries did not have constant access to a bank account, banking institution, mail, computer, or smartphone [43]. Beyond the unique vulnerabilities faced by individuals experiencing homelessness, individuals experiencing homelessness and a brain injury faced more barriers to accessing housing and financial resources during COVID-19.

*4.5. Limitations*

Certain study limitations should be noted. The study results are based on cross-sectional data, reducing the ability to draw causal conclusions. Future research on the impact of COVID-19 may consider longitudinal data to draw causal relationships. Additionally, the data used were self-reported and could be biased due to the sensitive topics asked of the participants. While the survey was designed not to cause any distress; the participants may have been uncomfortable. Further, the participants were recruited from service agencies for adults experiencing homelessness, so the sample is likely not representative of all adults experiencing homelessness, particularly those adults who are not seeking services. In addition, this study was limited geographically to one city in Colorado, and while the demographic information is largely reflective of the demographics of the city and state, additional research should consider using various geographic areas where there are more diverse populations. Furthermore, the qualitative data collected were designed to elicit mostly short answer responses. Researchers may consider an open-ended interview style data collection in the future to gain a better understanding of the lived experiences of individuals experiencing homelessness and TBI during COVID-19. Additionally, this study only included English-speaking adults. Researchers should consider administering their study in multiple languages in the future. Lastly, access to care is an issue for all people experiencing homelessness, not only individuals experiencing homelessness with a reported TBI; this study does not suggest there are different access issues, but rather, examines access issues specifically for a subpopulation of individuals experiencing homelessness with a reported TBI.

**5. Conclusions**

In the case that there is another global pandemic, there needs to be additional social systems in place to prevent further marginalization of people experiencing homelessness

and TBIs. Individuals experiencing homelessness and a TBI often rely on social services for food services, financial assistance, and a place of social connection; therefore, closed agencies and limited transportation can increase the risk of isolation. Maintaining social connections is important for people experiencing homelessness and living with a TBI because social support can help connect individuals to resources [37]. Rather than continue to invest in congregate shelters, funds should be allocated to more permanent supportive housing structures, tiny home villages, safe parking lots, and safe camping spaces. Investing funds into supportive systems that not only provide for physiological needs, but also psychological needs, will alleviate the additional difficulties experienced by those who have a TBI and experiencing homelessness.

**Author Contributions:** Conceptualization, S.C., B.B., E.G., K.G. and D.B.; Formal analysis, S.C., K.C., B.B., G.S. and S.C.R.; Investigation, S.C.; Methodology, S.C., K.C., B.B., G.S., E.G. and S.C.R.; Project administration, S.C., K.G. and D.B.; Supervision, K.G. and D.B.; Writing—original draft, S.C., K.C., B.B., S.C.R., G.S. and E.G.; Writing—review and editing, S.C., K.C., B.B., S.C.R., K.G. and D.B. All authors have read and agreed to the published version of the manuscript.

**Funding:** This research was funded by the University of Denver Professional Research Opportunity for Faculty and by Mindsource Brain Injury Network in Colorado.

**Institutional Review Board Statement:** The study was conducted in accordance with the University of Denver and approved by the Institutional Review Board of University of Denver (1521142-9 and 10/19/21).

**Informed Consent Statement:** Informed consent was obtained from all subjects involved in the study.

**Data Availability Statement:** Not applicable.

**Conflicts of Interest:** The authors declare no conflict of interest.

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
