# Peer review of "The Impact of COVID-19 on Access to Resources among Individuals Experiencing Homelessness and Traumatic Brain Injury"

_traumacare, doi:10.3390/traumacare3010004_

Round 1

Reviewer 1 Report

My review is attached as a PDF.

Reviewer 2 Report

This is a description of an evaluative study of the intersection and therefore challenges in healthcare provision and support of those with TBI as well as homelessness during the COVID-19 pandemic and a description of the challenges and failures that occurred. The report uses a small number of cases (N=38) from one Colorado city(USA).

The paper is well organized and written but the weakness only lays in the relatively small number of individuals studied.  In addition there was no discussion of immigration status of the individuals involved and whether they had publicly funded/privately funded or No funded healthcare during this time. The implication was that all were without health care funding.

Analysis of other associated physical disabilities might have been informative, and the conclusions about homelessness and the need for structural change and shelter provision seems intuitive but could there have been exploration of other networks of support that might be more fluid and portable ( Staff and system navigators/advocates and communication technology support also have been recommended from their findings.)_

Reviewer 3 Report

This study examines the relationship between TBI and homelessness and the impact of COVID-19 on access to resources among 38 homeless participants who had experienced a TBI.

Affiliations should be authors institutes, not emails.

Lines 15, 64, 320, 369, 371, 373: Change "compared to" to "compared with

Line 21: Check formatting. This sentence should be reworded to simplify and improve clarity.

Line 42: This section explains the mechanism and consequences of TBI. I suggest adding details of the classification of TBI as mild, moderate or severe as this is important to signs and symptoms as well as treatment and long term consequences. 

Line 62: Delete "compared to"

Line 97: "called out" or "called in"?

Line 100: HUD?

In the "Impact of COVID-19" section you did not mention vaccination rates among the homeless. I think this is an important number to include. Also, how did mis- and disinformation about COVID-19 and vaccinations affect homeless peoples compliance with health mandates and advice such as wearing masks, social distancing, vaccination etc?

Lines 204-205: Why was data collected on only one day? Could the researchers have included more participants with multiple visits to the sites?

Line 291: Delete "Out"

Line 310: Change "kicked out of" to "evicted from"

Table 1: Under "TBI Variables" it would be helpful to include the TBI classification data as I mentioned earlier.

In the Discussion section "TBI," again addressing the different classifications of TBI would help.

Some statistics on the number of homeless people in the area who caught COVID-19 and those who did not would help to quantify the effect in that community.

Round 2

Reviewer 1 Report

The authors have substantially addressed the requests for revisions. I think a greater effort could still be made to indicate policy recommendations drawn from the results and conclusions (and that this would be quite beneficial), particularly as the abstract mentions the need for such policies. There is also a typo on line 258 (page 6) "Self-report".

Reviewer 3 Report

I believe the Authors have addressed all concerns of the reviewers and the manuscript is now appropriate for publication. One final minor suggestion, line 212: Change (the Centre for..." to (Centre for..." 

Author Response

One final minor suggestion, line 212: Change (the Centre for..." to (Centre for..."- Thank you this has been updated